# Molecular and Clinical Data of Antimicrobial Resistance in Microorganisms Producing Bacteremia in a Multicentric Cohort of Patients with Cancer in a Latin American Country

**DOI:** 10.3390/microorganisms11020359

**Published:** 2023-01-31

**Authors:** Sergio Andrés Cruz-Vargas, Laura García-Muñoz, Sonia Isabel Cuervo-Maldonado, Carlos Arturo Álvarez-Moreno, Carlos Humberto Saavedra-Trujillo, José Camilo Álvarez-Rodríguez, Angélica Arango-Gutiérrez, Julio César Gómez-Rincón, Katherine García-Guzman, Aura Lucía Leal, Javier Garzón-Herazo, Samuel Martínez-Vernaza, Fredy Orlando Guevara, Leydy Paola Jiménez-Cetina, Liliana Marcela Mora, Sandra Yamile Saavedra, Jorge Alberto Cortés

**Affiliations:** 1Department of Internal Medicine, Universidad Nacional de Colombia, Sede Bogotá, Bogotá 111321, Colombia; 2Infectious Diseases Group, Instituto Nacional de Cancerología-ESE, Bogotá 111511, Colombia; 3Research Group in Cancer Infectious Diseases and Hematological Alterations (GREICAH), Bogotá 111321, Colombia; 4Clínica Universitaria Colombia, Bogota 111321, Colombia; 5Hospital Universitario Clínica San Rafael, Bogotá 110111, Colombia; 6Department of Microbiology, Universidad Nacional de Colombia, Bogotá 111321, Colombia; 7Infectious Diseases Unit, Hospital Universitario San Ignacio, Bogotá 110231, Colombia; 8Research Group in Infectious Diseases, Hospital Universitario San Ignacio, Pontificia Universidad Javeriana, Bogotá 110231, Colombia; 9Infectious Diseases Department, Clínica Reina Sofía, Bogotá 110121, Colombia; 10Microbiology Laboratory, Instituto Nacional de Cancerología-ESE, Bogotá 111511, Colombia; 11Diseases Unit, Hospital Universitario Nacional, Bogotá 111321, Colombia

**Keywords:** bacteremia, drug resistance, microbial, beta-lactamases, adults, neoplasm, infection

## Abstract

Patients with cancer have a higher risk of severe bacterial infections. This study aims to determine the frequency, susceptibility profiles, and resistance genes of bacterial species involved in bacteremia, as well as risk factors associated with mortality in cancer patients in Colombia. In this prospective multicenter cohort study of adult patients with cancer and bacteremia, susceptibility testing was performed and selected resistance genes were identified. A multivariate regression analysis was carried out for the identification of risk factors for mortality. In 195 patients, 206 microorganisms were isolated. Gram-negative bacteria were more frequently found, in 142 cases (68.9%): 67 *Escherichia coli* (32.5%), 36 *Klebsiella pneumoniae* (17.4%), and 21 *Pseudomonas aeruginosa* (10.1%), and 18 other Gram-negative isolates (8.7%). *Staphylococcus aureus* represented 12.4% (n = 25). Among the isolates, resistance to at least one antibiotic was identified in 63% of them. Genes coding for extended-spectrum beta-lactamases and carbapenemases, blaCTX-M and blaKPC, respectively, were commonly found. Mortality rate was 25.6% and it was lower in those with adequate empirical antibiotic treatment (22.0% vs. 45.2%, OR: 0.26, 95% CI: 0.1–0.63, in the multivariate model). In Colombia, in patients with cancer and bacteremia, bacteria have a high resistance profile to beta-lactams, with a high incidence of extended-spectrum beta-lactamases and carbapenemases. Adequate empirical treatment diminishes mortality, and empirical selection of treatment in this environment of high resistance is of key importance.

## 1. Introduction

The complications associated with cancer are multiple. Among them, bacterial infections stand out, which vary according to the type of neoplasia and the treatment used. One in four patients with post-chemotherapy severe febrile neutropenia develops bacteremia [1], with a negative effect on initiating a specific cancer treatment, prolonged hospital stays, morbidity, and mortality, particularly when inappropriate treatment is used [2]. Cancer patients represent a particularly complex group due to multiple risk factors for developing infectious complications. These include immunosuppression due to the underlying disease and its treatment [3], exposure to broad-spectrum antimicrobials, with the consequent selection of multidrug-resistant microorganisms, and morbidity and mortality due to septic shock [4].

Antimicrobial resistance is a public health problem. According to the WHO, it is one of the crucial issues of the twenty-first century, with a significant socioeconomic impact worldwide [5]. This problem is especially important among Enterobacterales, such as *Escherichia coli*, *Klebsiella* spp., *Enterobacter* spp., but also among isolates of *Pseudomonas aeruginosa*, in which the resistance to third-generation cephalosporins and carbapenems is specially problematic [6]. González et al. conducted a systematic review in 2014, which demonstrated an increase in antimicrobial resistance between 2006 and 2011 in Enterobacteriaceae to third-generation cephalosporin in third-level hospitals in Colombia [7]. The expected effect of the resistance is a lower probability of an appropriate empirical therapy, which affects the morbidity and mortality of these immunocompromised patients [2].

Currently, there is no information about the susceptibility profile and molecular mechanisms of resistance in Latin America, among cancer patients with bacterial infection. The aim of this work is to identify the resistance profiles and describe genes related to resistant microorganisms in cancer patients who present an episode of bacteremia in hospitals in Colombia.

## 2. Materials and Methods

### 2.1. Setting

Patients from six reference hospitals in the city of Bogotá, Colombia, were included: Instituto Nacional de Cancerología (INC), Clínica Universitaria Colombia (CUC), Hospital Universitario Nacional de Colombia (HUNC), Clínica Reina Sofía (CRS), Hospital Universitario Clínica San Rafael (HUSR), and Hospital Universitario San Ignacio (HUSI), during the period between 1 July 2018, and 31 July 2020. All these hospitals have oncology or hemato-oncology care units; one of them is public (INC) and all serve patients belonging to both public and private health care providers. They serve Colombian patients from Bogotá or other cities by referral.

### 2.2. Study Design

A prospective, multicenter, cohort study in adult patients was designed with those diagnosed with cancer of any tumor location and stage who developed a bacteremia confirmed by the laboratory of the participating institutions.

The inclusion criteria considered adult individuals of any race, with a confirmed cancer diagnosis, with or without treatment (chemotherapy, radiotherapy, surgical management). For microorganisms that are part of the microbiota of the skin, bacteremia was considered when there were at least two positive blood cultures for the same germ; similarly, at least one blood culture was considered for other Gram-positive microorganisms or Gram-negative ones. Patients with incomplete medical history information were excluded from the study.

### 2.3. Study Size and Sample

Sampling was non-probabilistic, sequential, and based on compliance with inclusion and exclusion criteria. The sample size was calculated considering an expected prevalence of 30% for *Staphylococcus aureus* (most frequent individual microorganism expected) and 60 to 70% Gram-negatives (as a group) in bacteremia among cancer patients, with a precision of 10%, a power of 90%, and an alpha error of 0.05 (two-tailed). A sample of 200 patients was estimated.

### 2.4. Definitions

Bacteremia is the infection of the bloodstream caused by bacteria. A systemic inflammatory response is understood as the presence of two or more of the following criteria: temperature greater than 38 °C or less than 36 °C; heart rate greater than 90 beats per minute; respiratory rate greater than 20 breaths per minute; or leukocyte count greater than 12,000 cells/mm^3^, less than 4000 cells/mm^3^, or more than 10% of bandemia [8]. Sepsis implies the presence of systemic inflammatory response, given that all patients had an identified infectious focus. Febrile neutropenia is an absolute count below 500 cells per milliliter or 1000 cells expected to fall below 500 cells per milliliter within 48 h. Additionally, it includes the presence of fever, defined as two consecutive temperature measurements of 38 °C for 2 h or an oral temperature greater than 38.3 °C [9]. Lymphopenia consists of a lymphocyte count of fewer than 1500 cells/mm^3^ and is severe with a count of less than 500 cells/mm^3^ [10].

Natural resistance profile is defined as the resistance pattern of each bacterial family, species, or group isolated in the antibiogram [11]. Acquired resistance profile, evidenced through the antibiogram, is characterized by not being typical of the bacterial family, species, or group; it is variable and obtained by a strain of a particular species [11]. Adequate antimicrobial treatment is antibiotic management with activity against the bacteria isolated in the blood culture, applied at the onset of symptoms or signs of infection, and maintained during the following 72 h. It is considered effective in vitro against the isolated microorganisms [12].

### 2.5. Microbiological Procedures

Isolates of the bacteria identified in the blood cultures of cancer patients who met the inclusion criteria and not the exclusion criteria were collected. Initial identification at the genus and species level and measurement of sensitivity to antimicrobials of the isolates were carried out in the clinical laboratory of the selected hospitals of Bogotá, with the automated systems of each institution: VITEK (INC, HUSI, CUC, CRS, Biomerieux, France) and Phoenix 100 (HUNC, HUSR, BD, USA), according to the manufacturers’ instructions. No molecular methods were used for identification. Antibiotics tested by every hospital were chosen by each center. Commonly tested antibiotics include ampicillin, cephalosporins, carbapenems, aminoglycosides (amikacin and gentamicin), ciprofloxacin, and trimethoprim/sulfametoxazole. Colistin was tested against *P. aeruginosa* isolates. Multidrug resistance (MDR) was defined as the presence of resistance to 3 or more antibiotic classes.

Two infectious disease specialists (JAC and SCM) reviewed, individually, the antibiograms and gave their conclusions according to the resistance phenotypic profiles of the microorganism evaluated. A third infectologist (JSB) evaluated the discordances that arose, and the final phenotypic profile was defined by agreement.

The isolates that showed some type of resistance as described in the study protocol were transferred in Cary–Blair medium to the Microbiology Laboratory of the Universidad Nacional de Colombia to be stored for subsequent molecular testing. Before the molecular study, the cryopreservation of strains was verified. The DNA of the isolates was obtained using cell lysis from fresh bacterial colonies of culture in non-selective agar. The sample was diluted at 1:10, taking 50 μL of the supernatant and placing it in a tube for DNA dilution (450 μL of sterile Type I water). This DNA sample was stored in a cold room when processed within three days; otherwise, in a −20 °C refrigerator until processing (the samples at −20 °C were thawed gradually by placing them in a cold room). Detection of resistance determinants was performed by conventional PCR. In Gram-negative bacilli, the presence of beta-lactamase-encoding genes was evaluated using specific primers for blaSHV, blaTEM, blaCTX-M [11], blaKPC, blaGES, blaVIM [12], blaNDM [13], blaIMP [14], and blaOXA-48 [15]. In Staphylococcus spp., resistance to methicillin was confirmed by detecting the mecA gene [16], and in Enterococcus spp., for vancomycin resistance, the presence of the vanA and vanB genes was evaluated [17]. DNA was added to tubes, vortexed, and 5 μL of each sample was placed in a thermocycler. A 1.2% agarose gel was prepared, and, finally, the electrophoresis was carried out.

### 2.6. Ethical Considerations

The study was conducted in accordance with good clinical practice (GCP) standards. The level of ethical risk at which the study is classified according to Article 11 of Resolution No. 008430 of 1993 issued by the Ministry of Health of Colombia is “Research with minimum risk”. To guarantee confidentiality, anonymous data were used. The protocol was submitted for review and approved by the Ethics Committee of each of the participating institutions.

### 2.7. Statistical Analysis

Data were entered into the REDCAP platform, and their statistical analysis was carried out using the R program (version 4.0.6). Demographic characteristics were recorded in relative and absolute frequencies, and measures of central tendency were calculated for continuous variables as mean (standard deviation) and median (interquartile range) according to their distribution. The exploratory analysis compared the proportions of qualitative variables using the X2 test, and a Yates correction was applied when there were fewer than 100 observations. The multivariate analysis used a logistic regression method to predict mortality, considering clinical variables and inappropriate antimicrobial therapy as predictors. Results with *p* < 0.05 were considered significant.

## 3. Results

### 3.1. Characteristics of the Population

In the 195 patients included, 206 microorganisms were identified: 120 (61.5%) from INC, 25 (12.8%) from CUC, 19 (9.7%) from HUSI, 16 (8.2%) from HUSR, 9 (4.6%) from HUNC, and 6 (3.1%) from CRS. There were 109 women (55.8%), and the average age was 52 years (SD 17.2 years). The majority (104, 53.3%) had hematological cancer. Of the total number of patients, 83 (42.5%) had active oncological pathology, 43 (22%) were in relapse, 41 (21%) had de novo cancer, 15 (7.7%) were in remission, and 13 (6.6%) were in palliative care for their pathology. As for cancer treatment, in the last month, 105 (53.8%) had received chemotherapy, 23 (11.8%) radiation therapy, and 49 (25.1%) some surgery. Only six (5.7%) of the patients with hematological cancer had a hematopoietic progenitor transplantation. Among the patients who received chemotherapy, 24 (22.8%) had an implantable catheter. Of the total patients included, 52 (26.6%) had had a previous infection, and 70 (35.8%) had received antibiotic treatment during the previous month. The most frequently used ones were non-carbapenem beta-lactams with 43 episodes (21.9%), followed by carbapenems with 18 episodes (9.2%), and vancomycin with 14 episodes (7.1%). At the time of presenting bacteremia, 131 (67.1%) were in hospital, 56 (28.7%) in the emergency room, and 8 (4.1%) in the intensive care unit; more than a half of them had a medical device (Table 1).

Severe lymphopenia was also identified in 111 patients (56.9%), hyperbilirubinemia in 43 (33.3%, tested in 129), and acidemia in 7 (8.8%, tested in 80), with significant metabolic acidosis/acidemia in 40% of these. Hypoxemia was observed in 37 cases (48.6%, measured in 76) and hyperlactatemia in 31 patients (37.3%, measured in 83), and acute kidney injury was found in 19 (10.4%).

### 3.2. Microbiological Results

In the 195 patients included, 206 microorganisms were identified; bacteremia was polymicrobial in 11 patients (5.6%). A total of 142 Gram-negative (68.9%) and 64 Gram-positive (31.1%) isolates were found (Table 2). The principal Gram-negative germ isolated was *Escherichia coli*, followed by *Klebsiella pneumoniae* and *Pseudomonas aeruginosa*. Among the Gram-positive bacteria, the main isolates correspond to coagulase-negative *Staphylococci*, followed by *S. aureus*, *Enterococcus* species, *Streptococcus* species, *Bacillus*, and *Lysinibacillus*. Most isolates, 130 (63.1%), had an acquired resistance pattern. In 192 patients (98.4%), initial antibiotic treatment was prescribed, which was adequate in 155 cases (79.5%).

Among Gram-negative bacteria, the phenotype observed in the resistance antibiograms of third- and fourth-generation cephalosporins was found in *E. coli* in 13 cases (19.7%), in 3 in *K. pneumoniae* (8.3%), and in 2 cases in *P. aeruginosa* (9.5%). No resistance to carbapenems was observed in *E. coli*, but it was found in 12 isolates of *K. pneumoniae* (34.2%) and 4 of *P. aeruginosa* (19%) (Table 3). Regarding the resistance of Gram-negative bacteria, in general, resistance to third- and fourth-generation cephalosporins was evidenced in 29 cases (22.4%) and to carbapenems in 15 (10.9%). MDR isolates were seen in E. coli (12, 17.4%), *K. pneumoniae* (16, 44.4%), *P. aeruginosa* (7, 33.3%), Serratia marcescens (1, 100%), and *S. maltophilia* (1, 100%). Among Gram-positives, MDR isolates were seen in *S. aureus* (7, 28%, all methicillin-resistant *S. aureus* isolates were classified as MDR), CNS (14, 53.8%), and *E. faecalis* (1, 25%)

Genes associated with resistance were found in 33 bacterial isolates studied: the principal gene was *blaCTX-M* in 14 cases, followed by *blaTEM* in 11, and *blaSHV* in 4 isolates of *E. coli* or *K. pneumoniae*. In two isolates, all three genes were found simultaneously. Among the carbapenem-resistant bacteria, the most common gene was *blaKPC* in 13 of 15 isolates (87%), followed by *blaVIM* in 2 (13%). No other causative genes, such as *blaGES*, *blaOXA-48*, *blaNDM*, or *blaIMP* (Table 4), were found; it is important to highlight that the analysis of one of the isolates of *P. aeruginosa* was not possible due to limited sample availability.

Among the Gram-positive microorganisms, six (24%) isolates of *S. aureus* resistant to oxacillin were observed, and none were resistant to vancomycin. In *E. faecalis*, no resistance to ampicillin or vancomycin was found. In *E. faecium*, four (100%) ampicillin-resistant and two (50%) vancomycin-resistant isolates were found. Of the isolates of *S. aureus* resistant to oxacillin, 83% had *mecA*. Among the vancomycin-resistant isolates of *E. faecium*, the *vanA* gene caused 100%.

A history of antibiotic use during the month before bacteremia was observed in 52 patients (40.9%) with microorganisms with an acquired profile (*n* = 127), and in 18 (26.4%) with microorganisms with a natural susceptibility profile (*n* = 68; OR: 1.91, 95% CI: 0.97–3.9). Among patients with Gram-negative *bacilli* isolates (*n* = 142), 22 (59.5%) had received a previous antibiotic from those identified as having a third- or fourth-generation cephalosporin-resistant or carbapenem-resistant microorganism (*n* = 37). This exposure occurred in 26 (24.8%) patients without this profile (*n* = 105; OR: 4.5, 95% CI: 2.0–9.8).

### 3.3. Mortality and Risk Factors

In the group of patients with resistant Gram-negative infections, four (22% of 18) with bacteria resistant to third- or fourth-generation cephalosporins and three (18.7% of 16) with bacteria resistant to carbapenems died at 30 days (*p* = 1.0). Two patients (33.3% of 6) in the group with MRSA bacteremia and two (100%) patients with vancomycin-resistant *E. faecium* died at 30 days (*p* = 0.42).

At day 30, from the time of identifying bacteremia, 134 (68.7%) patients had been discharged, 50 (25.6%) had died, and 11 (5.6%) were still hospitalized. Among the survivors, 56 (38.6%) patients had a natural profile, while 12 (24%) of those who died had this profile (OR: 0.50, 95% CI: 0.22–1.09, *p* = 0.09). Differences in mortality were observed according to the clinical stage of the patient: 0% mortality (0 of 15 patients) in patients in complete remission; 19.2% (16 of 83 patients) in those with active disease (*p* = 0.12); 23.8% (10 of 42) in those with de novo disease; 39.7% (17 of 43) in patients in relapse (*p* = 0.21); and 53.8% (7 of 13) for patients in palliative care (*p* < 0.01, compared to those in remission).

In the multivariate analysis, an inverse relationship was found between adequate treatment and mortality (OR: 0.26, 95% CI: 0.1–0.63), in addition to a relationship between mortality and palliative care (OR: 3.51, 95% CI: 1.05–12.04). Finally, a relationship was observed between survival and the number of platelets (per 10,000 additional platelets) (OR: 0.97, 95% CI: 0.94–0.99).

## 4. Discussion

Our study shows high antibiotic resistance among bacteremia-causing microorganisms in cancer patients in a developing country. Resistance rates were higher among Gram-negative bacteria and especially among *K. pneumoniae* isolates. In addition, it presents the characterization of the principal genes responsible for resistant bacteria, among which *blaCTX-M* and *blaKPC* stand out in the group of extended-spectrum beta-lactamases and carbapenems, respectively.

The microbiological findings showed a predominance of Gram-negative microorganisms in 68.6% of the isolates. These findings are in concordance with recent studies that evidence a predominance of this type of bacteria [18,19,20] and contrast with research carried out in the 2000s where the main bacteria causing infection in the bloodstream were Gram-positive bacteria [21]. It has been observed that the frequency of Gram-negative and Gram-positive microorganisms changes according to the geographical area [22,23], hence the importance of having regional information. Changes in epidemiology may be related to multiple causes, such as the utilization of antibiotic prophylaxis, campaigns to promote the reduction of device-associated bacteremia, the types of chemotherapy applied that have been more intensive during the last decade, or the lower use of chemotherapy catheters, among others [18,24].

Among the microorganisms identified, an acquired resistance profile was found in 63% of the isolates, in agreement with recent studies that have shown a gradual increase in antimicrobial resistance; among the proposed mechanisms, selective pressure due to inadequate use of antibiotics has been considered [22,25]. In South Korea, Nham et al. found a relationship between the development of BLEE-producing *K. pneumoniae* and the previous use of antimicrobials [20], without ruling out cross-bacterial transmission in different institutions, especially in those with low antimicrobial use, and the introduction of resistant bacteria from the community [26].

The most frequent resistance profiles varied according to the species identified. In *E. coli*, the most phenotypically and genotypically recognized resistance was due to BLEE; resistance to carbapenems was observed in *K. pneumoniae*, and multiresistance in *P. aeruginosa*. Upon genetic evaluation, 29 genes associated with bacteria resistant to third- and fourth-generation cephalosporins were found. The main gene involved was *blaCTX-M*, followed by *blaTEM* and finally *blaSHV*. In a group of neutropenic patients with hematological cancer and colonized by BLEE-producing *E. coli*¸ Arnan et al. (Spain) found *blaCTX-M* as the main gene associated with 81% of the isolates, followed by *blaSHV* (15%) and *blaTEM* (3.3%) [27]. In a group of patients with bacteremia and hematological cancer with BLEE-producing *E. coli* infection, Trecarichi et al. (Italy) described *blaTEM* as the main gene related to 36.1%, followed by *blaCTX-M* (33.3%) and *blaSHV* (30.6%) [28], which indicates that resistance patterns may be due to the geographical distribution of resistance genes. On the other hand, resistance to carbapenems was found in 34.2% and 19% of the *K. pneumoniae* and *P. aeruginosa* isolates, respectively. The main gene related to this resistance was *blaKPC*, followed by the *blaVIM* gene. Satlin et al. performed the characterization of carbapenem-resistant *Enterobacteriaceae* in patients with hematological cancer, neutropenia, and bacteremia in New York [29]. In this study, the main gene related to this resistance was *blaKPC* in 91% of the isolates. The high mortality rate of 53% at 30 days in the study by Satlin et al., compared to 18.7% in our study, associated with carbapenem-resistant bacteria, is noteworthy. Our data are consistent with what was observed in general among carbapenem-resistant isolates in Colombia, where the most frequently identified gene is *blaKPC*, regardless of the species [30]. We found a genetic variation according to the geographical area studied, which may be relevant when developing molecular diagnostic technology [31].

In patients with *S. aureus*, resistance to oxacillin was found in 24%; this is similar to what has been found in the international literature, which varies from 18–44% [32,33]. Regarding the result of 83% with evidence of the *mecA* gene as a mechanism of resistance, this is striking given it is the most commonly found gene, but other mechanisms not described to date in Colombia, such as *mecC*, can have the same resistance profile [34]. As for *E. faecium*, 50% resistance to vancomycin was found, 100% related to the *vanA* gene, with a mortality of 100% at 30 days. Observed mortality among patients with cancer infected with vancomycin-resistant *E. faecium* has been very high [35]. Vancomycin, piperacillin/tazobactam, meropenem, and metronidazole have been implicated in its appearance. It may cause bacteremia, but also complicated abdominal surgery, and skin-related infection. Long-term mortality has also been observed among patients with urological or colorectal malignancies [36]. Additionally, there is a considerable risk of outbreaks of these MDR microorganisms that might put a large number of patients in danger [37].

Regarding the outcomes, we did not find significant differences in relation to survival in patients with an acquired resistance profile compared to natural susceptibility. The information regarding this finding is contradictory. Tumbarello et al. in 2009 described a relationship between mortality and resistant bacteria [38]; among other factors related to mortality, age, comorbidities, and inappropriate use of antimicrobials have also been suggested [39].

A relationship was found between adequate initial antimicrobial treatment and survival, which is consistent with what has been described in multiple international studies that demonstrated a clear association between the use of inadequate antibiotic treatment and increased mortality [29,40,41,42]. Similarly, the group of patients in palliative care had a higher mortality rate, similar to what was previously described in the literature [18]. This finding, in relation to the prudent use of antimicrobials at the end of life, raises the need for interdisciplinary interventions to define the duration of antibiotics in this group of patients to reduce their impact on the microbiota [43]. Finally, thrombocytopenia was also associated with mortality in this group of patients. In 2012, Horino et al. described thrombocytopenia as a factor of poor prognosis in patients with *P. aeruginosa* bacteremia of which about 60% were cancer patients [44], which may represent poor control of the oncologic disease or an effect of septic shock.

Our study has some limitations, including the fact that despite being multicenter, most patients came from the INC because it is the largest public hospital in Colombia that treats cancer patients; the other participating institutions have oncology units in addition to treating other types of pathologies. When studying cancer patients, we might be analyzing a heterogeneous group due to the epidemiological and clinical differences of different types of oncologic pathologies.

On the other hand, this is the first Colombian study that reports phenotypic and molecular resistance patterns in patients with cancer and bacteremia. Considering that bacterial resistance is different in each geographic area, the results presented here are of great interest, given that they contribute to recommendations to strengthen programs aimed at the prudent use of antimicrobials and to encourage the creation or improved functioning of infection committees in institutions that treat cancer patients [45]. This information might also be helpful for the development and implementation of molecular diagnostic tests in the country to identify bacterial resistance genes as a useful tool in decision-making for sensible antimicrobial use [31].

## 5. Conclusions

In this cohort study in hospitals in a developing country, most of the bacteria identified were Gram-negative, with elevated resistance to beta-lactams with the production of ESBL-type beta-lactamases and carbapenemases, mediated by specific genes. The observed resistance profile is high and had an effect on the probability of administering an appropriate antibiotic therapy. Patients with bacteremia and palliative care had also a higher risk of dying by day 30.

## Figures and Tables

**Table 1 microorganisms-11-00359-t001:** Characteristics of the patients included.

Initial Clinical Variables and History	*n* = 195 *n* (%)
Age (years, average)	52 (SD 17.2)
Female	109 (55.8)
Type of tumor	
Solid tumor	91 (46.7)
Hematological tumor	104 (53.3)
Tumor status	
Active tumor	83 (42.5)
Tumor in palliative care	13 (6.6)
De novo tumor	41 (21)
Relapsed tumor	43 (22)
Tumor in remission	15 (7.7)
Treatment received	
Chemotherapy (%)	105 (53.8)
Radiation therapy (%)	23 (11.8)
Surgery (%)	49 (25.1)
Hematopoietic progenitor transplantation (%)	6 (5.7)
Previous bacterial infection (%)	52 (26.6)
Antibiotic during the previous month (%)	70 (35.8)
Non-carbapenem beta-lactams	43 (22)
Carbapenems	18 (9.2)
Glycopeptides	14 (7.1)
Quinolones	4 (2.0)
Aminoglycosides	2 (1)
Other	9 (4.6)
≥2 antibiotics	30 (15.3)
Steroid use	10 (5.1)
Patient service	
Hospitalization	131 (67.1)
Emergency room	56 (28.7)
Intensive care unit	8 (4.1)
Implantable catheter	24 (12.2%)
Device usage	112 (57.4)
Central venous catheter	69 (35.3)
Bladder catheter	18 (9.2)
Nasogastric tube	14 (7.1)
Drain	10 (5.1)
Hemodialysis catheter	3 (1.5)
Other	32 (16.4)
Comorbidities	
Diabetes mellitus	17 (8.7)
Diabetes complications	1 (0.5)
Chronic kidney disease	16 (8.2)
Charlson index (median, IQR)	3 (2–3)
Systolic blood pressure mm Hg (mean, SD)	110 (21)
Diastolic blood pressure mm Hg (mean, SD)	65 (13)
Heart rate (mean, SD)	102 (20)
Temperature °C (mean, SD)	37.5 (1.2)
Respiratory rate (mean, SD)	20 (2)
Neutropenia	63 (32.3)
Febrile neutropenia	59 (30.2)
Systemic inflammatory response and sepsis	119 (61)
Fever	78 (40)

**Table 2 microorganisms-11-00359-t002:** Microbiological results of 195 patients with cancer and bacteremia.

Microbiological Results	*n* = 206 (%)
Gram-negative	142 (68.9)
*Escherichia coli*	67 (32.5)
*Klebsiella pneumoniae*	36 (17.4)
*Pseudomonas aeruginosa*	21 (10.1)
Other *Enterobacteriaceae*	15 (7.2)
*Acinetobacter*	2 (1)
*Stenotrophomonas maltophilia*	1 (0.5)
Gram-positive	64 (31)
*Staphylococcus aureus*	25 (12.4)
Coagulase-negative *Staphylococci*	26 (12.6)
*Enterococcus faecium*	4 (1.9)
*Enterococcus faecalis*	4 (1.9)
Other Gram-positive	5 (2.4)
Natural profile (%)	76 (36.9)
Acquired profile (%)	130 (63.1)

**Table 3 microorganisms-11-00359-t003:** Percentage of resistance of selected Gram-negative bacteria.

Antibiotic	*E. coli* *n* = 67	*K. pneumoniae* *n* = 36	*P. aeruginosa* *n* = 21
Ampicillin R/*n* (%R)	42/57 (73)		
Ampicillin/sulbactam R/*n* (%R)	11/65 (16.4)	18/35 (51.4)	
Cefazolin R/*n* (%R)	8/52 (15.3)	6/21 (28.5)	
Piperacillin/tazobactam R/*n* (%R)	2/65 (3.1)	14/35 (40)	1/17 (5,8)
Ceftazidime R/*n* (%R)	13/66 (19.7)	15/35 (42.8)	5/17 (29.4)
Ceftriaxone R/*n* (%R)	13/66 (19.7)	15/35 (42.8)	
Cefepime R/*n* (%R)	13/66 (19.7)	15/35 (42.8)	5/21 (23.8)
Ertapenem R/*n* (%R)	0	12/35 (34.2)	
Meropenem R/*n* (%R)	0	12/35 (34.2)	4/21 (19)
Amikacin R/*n* (%R)	0	1/36 (2.7)	3/21 (14.2)
Gentamicin R/*n* (%R)	10/64 (15.6)	5/34 (14.7)	3/21 (14.2)
Ciprofloxacin R/*n* (%R)	24/67 (35.8)	6/35 (17.1)	2/21 (9.5)
Trimethoprim–ulfamethoxazole R/*n* (%R)	38/59 (64.4)	11/23 (47.8)	
Colistin R/*n* (%R)			0

R/*n*: antibiotic-resistant bacteria/antibiotic-tested isolates.

**Table 4 microorganisms-11-00359-t004:** Resistance genes identified in Gram-negative *bacilli* isolates in cancer patients according to species.

Bacterium	Gene Identified	Total
*E. coli*	*blaCTX-M*	4
	*blaCTX-M, blaTEM*	7
	*blaCTX-M, blaTEM, blaSHV*	1
*K. pneumoniae*	*blaTEM*	1
	*blaSHV*	1
	*blaCTX-M, blaTEM*	1
	*blaCTX-M, blaTEM, blaSHV*	1
	*blaKPC*	11
	*blaVIM*	1
*P. aeruginosa*	*blaSHV*	1
	*blaKPC*	2
	*blaVIM*	1

## Data Availability

Not applicable.

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
