# Peer review of "Molecular and Clinical Data of Antimicrobial Resistance in Microorganisms Producing Bacteremia in a Multicentric Cohort of Patients with Cancer in a Latin American Country"

_microorganisms, 2023, doi:10.3390/microorganisms11020359_

Round 1
Reviewer 1 Report
Dear authors
Please, find the attached file.

Author Response
Line 39: Replace by bacterial species
Answer: Replaced.
Line 40: Rephrase as bacterial isolates not collected.
Answer: Rephrased.
Line 42 : On which basis the authors select resistance genes.
Answer: They were selected as these genes correspond to those more frequently found accounting for resistance to third generations cephalosporines and carbapenems in Colombia (added in line 152, methods).
Line 45: Rephrase this sentence to be clear.
Answer: It was changed: Gram-negative bacteria were more frequently found, in 142 cases (68.9%): 67 Escherichia coli (32.5%), 36 Klebsiella pneumoniae (17.4%), and 21 Pseudomonas aeruginosa (10.1%), and 18 other Gram negative isolates (8.7%).
Line 46: What is meant by resistance here?
Answer. The following was added to aid to the meaning: “Among the isolates, resistance to at least one antibiotic was identified in 63% of them”.
Line 49: Add precise percentages
Answer: They were added.
Line 49: For each isolates? correlate.
Answer: The odds ratio were the estimator in the final multivariate model. It was explained.
Line 57: Talk in brief about the most dangerous one and their negative impacts.
Answer: Gram negatives and their importance were added in the second paragraph (lines 77-85), together with references.
Lines 76-77: The introduction stated all the points related to the subject of study without referring to the problem that the authors discuss regarding the point that the authors start to complete the end of previous studies. Talk in details about the aim of the study.
Answer: The paragrapha was rewrite to describe the aim of the study (lines 86-89).
Line 80: The company names for any reagents must be mentioned in the materials and methods section.
Answer: They were added in the appropriate places.
Line 89: Write about the details of the patients examined
Answer: The characteristics and inclusion criteria are detailed in the 2.2. section, study details (lines 100-117)
Line 103: What about other bacterial species.
Answer: Staphylococcus aureus is the most frequently identified species in the cancer hospital (31%, Enferm Infecc Microbiol Clin. 2010;28(6):349–354). It also means around 60-70% of Gram negatives. So it was used as a reference for the sample size calculation.
Line 111: What about neutrophils?
Answer: The bandemia criteria was added (line 131).
Line 129: Talk in details about the identification of bacterial species.
What about the molecular identification?
Answer: Identification was done with the use of the automated systems as described. A recent paper has shown an accuracy of 98.6% for the most frequently used system (Vitek, Clin Lab. 2022 Oct 1;68(10). doi: 10.7754/Clin.Lab.2022.211247.). It was explained that fabricant instructions were followed (line 154).
Line 130: genus not gender.
Answer: It was corrected (line150)
Line 137: What is meant by meeting define susceptibility?
Answer: It meant that the phenotypic resistance profile was interpreted in consensus.
Line 149: On which basis the authors selected these genes?
Answer: The genes were selected form previous studies (Distribution and molecular characterization of beta-lactamases in Gram-negative bacteria in Colombia, 2001-2016. Rada AM, Biomedica. 2019 May 1;39(s1):199-220. doi: 10.7705/biomedica.v39i3.4351 and Biomedica. 2017 Dec 1;37(4):473-485. doi: 10.7705/biomedica.v37i4.3432). The references were added (lines 179-180).
Line 149: What are the primer sequences for these genes or refer to papers.
Answer: References were added (line 184)).
Line 152: Where are the amplification procedures and cycling conditions?
Answer: They were referred to the references (line 184) .
Line 174: Where are the P values for all results?
Answer: Since it is a descriptive study of resistance, no p values were calculated. In the analytic part, estimators used were Odds Ratios and confidence interval were added. Since they provide more information that the p value alone, we did not added p values for those calculations.
Line 176: Rephrase this sentence.
Answer: It was rephrased: “In the 195 patients included, 206 microorganisms were identified(line 207)
Line 218 (table 3): The antibiotics did not be mentioned in the materials section.
Answer: They were added in the methods section (lines 154-157)
Line 230: Add MDR percentages for each species.
Answer: A definition of MDR was added (lines 157-160) in the methods section. The number and percentage were added in lines 251-254..
Line 230: these genes did not be mentioned in the materials section.
Answer: The information was added in line 181-182.
Line 243: Not precise title.
Answer: The title was changed to “Mortality and risk factors” (line 280)
Line 313: All bacterial species must be italic.
Answer: The text was revised in its entirely (line 350).
Line 321: Add the danger of dissemination of VRSA.
Answer: It was added (line 359-361)
Line 335: interpret.
Answer: An interpretation was added (line 421-422)
Line 355: Mention the most important findings for this study in this section.
Answer: Other important results were added to this paragraph (lines 443-445).
Line 371: Add more recent references.
Answer: Some more recent references were added: 19, 20, 21, 22. An old reference (21- Jones et al) was deleted.
Reviewer 2 Report
Some minor notes
(1) Line 136: A third infectologist (JSB) evaluated the discordances…
Who is JSB? I could not find him (her) among the authors.
(2) Line 249: This At day 30, from the time of identifying..
“This” should be removed
(3) Line 317: As for [6][5][7][6]E. faecium, 50% resistance
Line 331: … and increased mortality [5][10][9][11][10][12][11]. Similarly..
Line 337-338: … and increased mortality [5][10][9][11][10][12][11]. Similarly…
What are these numbers is brackets? Please, correct.
Author Response
Some minor notes
(1) Line 136: A third infectologist (JSB) evaluated the discordances…
Who is JSB? I could not find him (her) among the authors.
Answer: Juan Sebastián Bravo is an infectious diseases specialist, outside of study, that helped in that task. His name was included in the aknowledgment.
(2) Line 249: This At day 30, from the time of identifying..
“This” should be removed
Answer: It was removed.
(3) Line 317: As for [6][5][7][6]E. faecium, 50% resistance
Line 331: … and increased mortality [5][10][9][11][10][12][11]. Similarly..
Line 337-338: … and increased mortality [5][10][9][11][10][12][11]. Similarly…
What are these numbers is brackets? Please, correct.
Answer: They were misspelling from references and versions. They were deleted, and the references adjusted.
Round 2
Reviewer 1 Report
Dear authors
Please, find the attached file

Author Response
Thank you for your kindly observations.
The following changes were done:
Line 81: Where is the gap for this study?
Answer: The text was changed as: “Currently, there is no information about the susceptibility profile and molecular mechanisms of resistance in Latin America, among cancer patients with bacterial infection.” (line 80-82).
Line 95: Write about the details of the patients examined
Answer: Information of the patients included were highlighted and explained (lines 91-94, 105-106, 108-110).
Line 112: What about other bacterial species. Please, clarify
Answer: Since 30% expected S. aureus, 60-70 % pf Gram negatives (as a group) was expected. Explained in lines 119-120.
Line 138: Talk in details about the identification of bacterial species.
What about the molecular identification?
Answer: No molecular identification was done. Bacterial genus and species wer identified thorugh the automated or semiautomated systems in each hospital. Explained in line 152-153.
Line 147: This is a result not materials.
Answer: It was moved to the results, added in line 278.
Line 150: What is meant by meeting define susceptibility?
Answer: It refered to an agreement to define the phenotypic resistance profile.
Line 170: What is meant by elsewhere? and where are the amplification procedures and cycling conditions?
Answer: The precise references to each one of the PCR procedures were added to the text (line 183-189).
Line 217: Where are the P values for all results?
Answer: When comparisons were made, a p value was added: line 309, 311, 315, 317-319.
Line 343: Focus on VRSA.
Answer: Information on this microorganism has been detailed in lines 381-386.
Line 378: Mention the most important findings for this study in this section rather than pointing general terms.
Answer: The paragrapha was chenged (lines 428-434)-
Line 400: Add more recent references.
Answer: Updated references included numbers 23, 24, 25,26, 30, 35,36,37,42,43